# Differential Response of Wheat Rhizosphere Bacterial Community to Plant Variety and Fertilization

**DOI:** 10.3390/ijms23073616

**Published:** 2022-03-25

**Authors:** Lisa Cangioli, Marco Mancini, Marco Napoli, Camilla Fagorzi, Simone Orlandini, Francesca Vaccaro, Alessio Mengoni

**Affiliations:** 1Department of Biology, University of Florence, Via Madonna del Piano 6, 50019 Sesto Fiorentino, Italy; lisa.cangioli@unifi.it (L.C.); camilla.fagorzi@unifi.it (C.F.); francesca.vaccaro@unifi.it (F.V.); 2Department of Agriculture, Food, Environment and Forestry (DAGRI), University of Florence, Piazzale delle Cascine 18, 50144 Florence, Italy; marco.mancini@unifi.it (M.M.); simone.orlandini@unifi.it (S.O.)

**Keywords:** wheat, rhizosphere, bacterial community, ecosystem functions

## Abstract

The taxonomic assemblage and functions of the plant bacterial community are strongly influenced by soil and host plant genotype. Crop breeding, especially after the massive use of nitrogen fertilizers which led to varieties responding better to nitrogen fertilization, has implicitly modified the ability of the plant root to recruit an effective bacterial community. Among the priorities for harnessing the plant bacterial community, plant genotype-by-microbiome interactions are stirring attention. Here, we analyzed the effect of plant variety and fertilization on the rhizosphere bacterial community. In particular, we clarified the presence in the bacterial community of a varietal effect of N and P fertilization treatment. 16S rRNA gene amplicon sequence analysis of rhizospheric soil, collected from four wheat varieties grown under four N-P fertilization regimes, and quantification of functional bacterial genes involved in the nitrogen cycle (*nifH*; *amoA*; *nirK* and *nosZ*) were performed. Results showed that variety played the most important role and that treatments did not affect either bacterial community diversity or bacterial phyla abundance. Variety-specific response of rhizosphere bacterial community was detected, both in relation to taxa (Nitrospira) and metabolic functions. In particular, the changes related to amino acid and aerobic metabolism and abundance of genes involved in the nitrogen cycle (*amoA* and *nosZ*), suggested that plant variety may lead to functional changes in the cycling of the plant-assimilable nitrogen.

## 1. Introduction

The plant bacterial community, and especially the rhizospheric bacterial community, has been clearly related to crop yield and resistance to biotic and abiotic stresses [1,2,3,4,5,6]. Domestication and breeding have strongly influenced plant bacterial community [7,8], resulting in some cases in a reduced ability of the plant to attract an adequate rhizosphere bacterial community [9]. Since the diversity of the plant bacterial community is contributing to the resilience and sustainability of agroecosystems to multiple stressors, including climate change, increased diversity of the crop-associated bacterial community can contribute to conjugating sustainable yield with a reduced footprint of agriculture on the environment. Indeed, in general terms, plant genotype is a relevant factor affecting diversity and functions of the plant bacterial community and influencing soil bacterial community [3,10,11,12]. Consequently, there is a growing focus on considering the recruited plant bacterial community for future crop breeding programs [3]. Among the priorities for harnessing the plant bacterial community, plant genotype × microbiome × environment × management interactions are stirring attention [13]. In fact, not only the species, but also the individuality of genotypes, viz. germplasm or varieties, show peculiarities in the associated bacterial community stresses [5,8,14].

With the increasing food demand [15,16], winter wheat grain production needs to expand by 11% by 2026 [17]. To increase the grain yield and protein production per hectare, intensive breeding of winter wheat started in the second half of the twentieth century [18]. Winter wheat cultivars before the late 1960s are commonly called “old”, while those registered later are called “modern” [19]. In particular, the introduction of growth repressor (Rht) genes led to the development of semi-dwarf genotypes [20] which are characterized by improved efficiency of partitioning photosynthetic assimilates and lodging resistance [18,21], are highly productive, homogeneous, with high protein production per hectare, and with a gluten composition suitable for industrial processing [19,22,23,24]. On the other hand, the intensive breeding of winter wheat led to lower vitamin and micronutrient concentrations in grains [25] as a consequence of the “yield dilution phenomenon” in “modern” varieties [26]. Furthermore, the smaller root system induced by the dwarfism gene [27] makes the “modern” varieties less able to explore the soil and therefore more reliant on chemical fertilization than the “old” ones [28]. Further, the breeding process may also have influenced the characteristics and quantity of root exudates in “modern” varieties, thus influencing the composition of the rhizosphere [20]. These features suggest that “old” and “modern” wheat varieties may harbor different root-associated microbiota related to the different nutrient requirements and root systems. In fact, concerning wheat-associated bacterial communities, a clear effect of variety on the rhizosphere bacterial community has been shown [29,30,31], as well as nitrogen fertilization [20,32]. Recently, a hypothesis of reduction of the level of plant-bacterial community interaction along wheat domestication has been proposed [33,34]. In this context, bacterial communities in the rhizosphere of “old” wheat varieties were found to be more taxonomically diverse than that detected for “modern wheat”, probably due to differences in root exudates [35].

Given the resurged importance of old wheat varieties as a source of germplasm for breeding to develop varieties with higher nutritional value and better use of soil nutrients, here, we wanted to identify in a panel of three old and one modern variety if the wheat rhizosphere bacterial community was more affected by the variety or by the fertilization. In particular, we aimed to investigate the presence of a differential effect that the same N and P fertilization treatment may have on different varieties. We answered this question by performing 16S rRNA gene amplicon sequencing analysis of rhizospheric soil, collected from three old and one modern wheat varieties grown under four N-P fertilization regimes. Taxonomic profiling and putative functional attribution were produced, as well as quantification of functional bacterial genes involved in the nitrogen cycle (nitrogen fixation, *nifH*; nitrification, *amoA*; denitrification, *nirK* and *nosZ*), to estimate the combined effect of wheat variety and fertilization on rhizosphere bacterial community ecosystem functions.

## 2. Results

### 2.1. Physicochemical Characterization of the Soil

Results of the physicochemical analysis of the soil samples are reported in Table 1. For each plot, the analysis was performed on bulk soil samples. A Principal Component Analysis (PCA) was performed on soil physicochemical parameters and the location of sample collection (i.e., latitude and longitude of the plots). No evident clustering was detected, either with respect to physicochemical parameters and geographical coordinates (Figure 1).

### 2.2. Diversity of the Soil and Rhizosphere Bacterial Community

A total of 3,180,595 16S rRNA reads were obtained, and 2,131,486 (67% of total reads) passed quality filtering (Appendix A). After the clustering step, a total of 12,795 amplicon sequence variants (ASVs) were obtained (Appendix A). To Bacteria, 10,271 ASVs were assigned, 59 ASVs to Archaea, and 2,465 were unassigned (NA). ASVs assigned to Archaea and NA were removed from the following analyses. Rarefaction curves obtained from ASVs reached plateaus for all samples (Appendix A), indicating a satisfactory survey of the bacterial diversity (Good’s coverage Appendix A), which allowed to estimate alpha diversity indices (Appendix A). Bulk soil and rhizosphere samples did not differ in alpha diversity nor in relation to both variety and fertilization (Permanova *p* > 0.05). Varieties differed in relation to the Shannon index, but not in relation to the Simpson index (Figure 2, see specific pairwise comparison for *p*-values), indicating that species richness and species abundance distribution (equitability) changed among varieties, while the dominance of bacterial groups did not.

Concerning the taxonomic diversity among samples, alpha and beta diversities of soil and rhizosphere samples were not separated (Figure 3), while plant varieties were, with evident grouping for Andriolo and Bologna, and less evident, but significant for Verna and Sieve (Appendix A).

Fertilization did not affect the taxonomic diversity of the samples. Statical analysis was performed using a permutational multivariate analysis of variance (PERMANOVA) on centered log-ratio-transformed samples (Appendix A).

### 2.3. Variety Affects the Taxonomic Composition of Wheat Rhizobacterial Community

Figure 4 reports the relative abundances of taxonomies at the phylum level for the top 20 ASVs detected (which represent relative abundances from 0.1% to 0.08%). Taxonomic abundances in relation to variety and fertilization are shown in Appendix A. The most represented phyla were Acidobacteriota, Actinobacteriota, Chloroflexi, Proteobacteria, and Verrucomicrobia. Differences among varieties were present in relation to phyla abundance (Figure 5). In fact, differential abundance analyses showed remarkable differences in Actinobacteriota, Chloroflexi, Desulfobacterota, Entotheonellaeota, Firmicutes, Myxococcota, Nitrospirota, and the candidate division RCP2-54, in particular for Verna variety, as confirmed by a Permanova on the PCA to the centered log-ratio (Appendix A). When considering single ASVs, 37 of them were found with statistically different abundance in relation to variety (Appendix A). These ASVs included members of the orders Burkholderiales, Rhizobiales (Beijerinckiaceae, Hyphomicrobiaceae, Xanthobacteraceae), Blastocatellales, Pyrinomonadales, as well as of Actinobacteriota.

### 2.4. Prediction of Potential Functions and Metabolic Pathways

Functional profiles of the bacterial community were inferred. All the identified metabolic pathways present in the taxa retrieved in the bacterial community are listed in Appendix A. We identified a total of 422 pathways, all distributed in all samples. However, pathways’ abundance showed a clustering according to varieties (Figure 6a). Using a SIMPER test, we then selected the top three pathways that mostly differentiate the four varieties (in terms of amount of variance explained). Six different pathways-(VALSYN-PWY, ILEUSYN-PWY, BRANCHED-CHAIN-AA-SYN-PWY, PWY-3001, PWY-3781, and PWY-7111) were selected, most of them were related to the metabolism of amino acids and aerobic respiration. In particular, the pathways of L-valine biosynthesis (VALSYN-PWY), L-isoleucine biosynthesis (ILEUSYN-PWY and PWY-3001) and aerobic respiration (PWY-3781) were found. Interestingly, this subset of pathways also suggested a clustering related to, in particular, Verna being grouped apart from the other three varieties (Figure 6b). Pathways related to the nitrogen cycle showed only a minor contribution to the overall differences (i.e., the denitrification pathway accounted to 0.05% of total variance).

### 2.5. Quantitative PCR Analysis of the Nitrogen Cycle Genes

In order to overcome the limitation of PICRUSt-based pathway predictions (which are based on overall pathways and on phylogenetic reconstruction) and more clearly define the effect of plant variety and fertilization on the bacterial community fraction involved in the nitrogen cycle, bacterial genes related to nitrogen fixation (from N_2_ to NH_3_, *nifH*), nitrification (conversion of ammonia NH_3_to hydroxylamine, *amoA*), and denitrification (nitrite to nitric oxide, *nirK* and nitrous oxide to N_2_, *nosZ*) were detected, and their abundance was estimated by quantitative PCR on rhizosphere samples (Figure 7). While for *nifH* no differences among varieties or fertilizations were detected, *amoA*, *nirK*, and *nosZ* were different with respect to varieties (Scott–Knott test *p* < 0.05, Appendix A). In particular, Andriolo and Bologna showed higher abundance of *nirK* and *amoA* genes than Sieve and Verna did. Verna also had lower *nosZ* values than the other three varieties did, suggesting that the denitrification pathway could be less effective on the rhizosphere of this variety.

## 3. Discussion

The taxonomic assemblage and functions of the plant bacterial community are strongly influenced by soil and host plant genotype [5,7,8]. Crop breeding, especially after the massive use of nitrogen fertilizers which led to varieties responding better to nitrogen fertilization [36], has implicitly decreased the pressure toward plant genotypes highly efficient in the use of native soil nutritional resources, then indirectly reducing the ability of the plant root to recruit an effective bacterial community [7,8,37]. Tough physicochemical analysis of soil did not detect evident groupings according to wheat varieties (apart from Andriolo), and the analysis of the bacterial community did, clearly differentiating the four varieties under study (Andriolo being the most heterogeneous, Figure 3). In fact, the overall body of our results showed that variety played the most important role and that fertilization did not affect either bacterial community diversity or bacterial phyla abundance. Interestingly, rhizosphere bacterial community and bulk soil bacterial community were highly similar, suggesting that under the plant density and the depth we sampled, the root apparatus could extend its effects at a certain distance from the plant stem (30 cm). This bacterial community was in agreement with the extensive root apparatus of wheat, which allows affecting the soil at a distance from the plant [38]. Consequently, we may consider “bona fide” our bulk soil samples as very close to a low-bound rhizosphere soil. It would be interesting to evaluate the extent in space around the plant of this assimilation of soil on the rhizosphere bacterial community, which can have relevance for the setting-up and interpretation of results from intercropping trials [39]. The three old varieties and the modern Bologna variety showed a differential abundance of several microbial taxa. Among them, Acidobacteriota, Bacteroidota, Chloroflexi, Firmicutes, and Nitrospirota were particularly relevant. Previous studies on wheat rhizosphere showed that N fertilization decreased the abundance of Acidobacteria and increased Bacteroidetes [20]. In our work, a decrease of Acidobacteriota was shown by Andriolo variety in the plots with increased P-fertilization (Appendix A), which may suggest that this variety may respond differently to P fertilization, in turn affecting its rhizosphere bacterial community. However, both Andriolo and Bologna showed levels of abundance of members of phyla Acidobacteriota and Bacteroidota higher than that of Sieve and Verna varieties. Since both Sieve and Verna derive from breeds with “Est Mottin 72”, we could not a priori exclude that the lower levels of Acidobacteriota and Bacteroidota in their rhizosphere may reflect this partially common genetic basis, highlighting a contribution of plant genotype to the interaction with specific bacterial phyla. If confirmed in following experiments, these bacterial phyla could be good candidates for quantitative trait loci (QTL) analysis identifying in breeds from the old Verna variety and the modern Bologna. The high abundance of members of Acidobacteriota also agrees with previous reports (see, for instance, [33,40]). Moreover, Bacteroidota, Chloroflexi, and Firmicutes phyla were recognized as important in differentiating wheat species (*T. durum* Desf. and *T. aestivum* L.) [33] and soil-varieties combinations [31], suggesting the presence of loci in the wheat genome which can be identified for interaction with the rhizosphere microbiota. To date, few loci/genes in a relatively small number of plant species have been identified as related to the recruitment of microbiota [41], and no studies have been performed on wheat. Interestingly, ASVs affiliated to the uncultured phylum RCP2-54 were found more abundantly in Sieve and mainly in Verna, compared to Bologna and Andriolo. The functional role of this phylum is still unknown; however, previous works showed its presence in the rhizosphere and in differentiating tomato and maize varieties under water deficit conditions [42], again suggesting a possible genetic basis in the plant for its differential recruitment. At the level of ASVs it was relevant to find members of Blastocatellales and Pyrinomonadales. A recent paper investigating soil suppressiveness to *Fusarium culmorum* in wheat, found Blastocatellales in the network associated with pathogen suppression [43], and Pyrinomonadales were reported to vary in abundance in wheat rhizosphere in relation to soil nitrate level [44]. This evidence may suggest that the four varieties in our study may be able to differentially recruit a fraction of the soil bacterial community able to improve plant health and contributing differently to plant use of nitrogen. Bologna variety had the highest levels of both Blastocatellales and Pyrinomonadales, although high variation between plots was present (Appendix A). The role of these bacterial groups on plant tolerance to pathogenic fungi and nitrogen use may deserve further attention.

Under the light of possible functional differences among the bacterial community recruited by the four plant varieties, PICRUSt2 results (Figure 6) mirrored in part the taxonomic profiling in varieties differentiation. In particular, Verna grouped separately from the other three varieties, similarly to the pattern shown in the nMDS analysis of ASVs (Figure 3). Interestingly, the metabolic pathways which mostly differentiated Verna from the other varieties were related to aerobic respiration and amino acid biosynthesis. We may speculate that root exudates from the four varieties differed in the amino acid composition, hence differentially promoting the growth of bacteria harboring specific biosynthesis pathways. Indeed, root exudates are rich in amino acids, and previous work showed that amino acid content strongly varies among different plant varieties. Concerning the aerobic respiration pathway, Verna showed a more abundant representation than the other varieties did, though with differences among fertilization regimes. This may suggest that the other varieties could hold a higher fraction of anaerobic bacteria/anaerobic metabolism. Verna rhizosphere bacterial community contained a higher amount of Actinobacteriota and Firmicutes, phyla including several aerobic bacterial groups thriving in soil, as Streptomyces and Bacillus.

Concerning possible differences in nitrogen cycling by the rhizosphere bacterial community (which is partially related to anaerobic metabolism), variation in the abundance of members of Nitrospirota was found in Verna and Sieve varieties compared to that in Bologna and Andriolo varieties. Nitrospirota includes Nitrospira, a well-known genus of nitrite oxidizers present in agricultural soil and in the plant rhizosphere [45]. Interestingly, this result agrees with qPCR estimation of functional gene abundance. In fact, the analysis of genes involved in the nitrogen cycle showed that *amoA* (nitrification) and *nosZ* (denitrification) were differentially abundant with respect to varieties, suggesting that variety may lead to functional changes in the cycling of the plant-assimilable nitrogen. However, we could not a priori exclude fertilization may partially contribute to this difference. Experiments with bulk soil samples taken at higher distances from the plant may be helpful to clarify how much of the differences resulted from the sole fertilization. Indeed, it would highly relevant from the perspective of low-input precision agriculture to clarify the extent and impact on plant productivity of such variety–fertilization interaction in the rhizosphere bacterial community, to exploit the potentialities of variety-specific bacterial community. Multiple seasons and long-term field experiments in different soil and climatic conditions would be needed to fully address this question.

## 4. Materials and Methods

### 4.1. Experimental Field and Plant Genotypes

An experimental field was established in October 2019 at the “Grappi Luchino” organic farm (43.0507° N, 11.6912° E) close to Pienza (Tuscany, Italy). The soil was a moderately alkaline (pH 8.2) silty clay, with 10.5% sand, 49% silt, and 40.5% clay. The 0–40 cm layer (the most affected by either plant root apparatus and agronomic practices) contained 131 (3.1) g kg^−1^ total lime, 9.3 (0.2) g kg^−1^ total organic carbon, 1119 (10) mg kg^−1^ total nitrogen (N), 5.5 (0.3) mg kg^−1^ available phosphorus (P), and 189.2 (6) mg kg^−1^ available potassium (K). Cultivation of crops was performed under an organic farming system. Chickpea (*Cicer arietinum*, L.) was the previous crop. Residues from the previous crop were used as manure.

Four Italian varieties of winter wheat (*Triticum aestivum*, L.) were used in this study, including three “old” genotypes (namely Andriolo, Sieve, and Verna) [19,24,28], and one dwarf registered cultivar (namely Bologna). According to Migliorini et al., (2016), the germplasm of the four varieties has different origins: Verna is an awn-less cultivar established in 1953 from a breed between “Est Mottin 72” × “Mont Calme 245” [28]. Sieve is an awn-less cultivar constituted in 1966 from “Est Mottin 72” × “Bellevue II” [46]. Andriolo is a Tuscan landrace having an unknown pedigree and dating back to the early 1900 [28]. Bologna is one of the most cultivated modern varieties in Italy owing to its quality and high yield stability; Bologna was bred in 1999 from (“H89092” × “H89136”) × “Soissons” by Società Italiana Sementi (Italy) [47].

The experiment included 16 treatments (parcels), which were combinations of four varieties of winter wheat, two nitrogen (N) fertilization levels (40 and 80 kg N ha^−1^ yr^−1^, N40 and N80, respectively), and two nitrogen-to-phosphoric anhydride (N:P_2_O_5_) fertilization ratios (NP) (N:P2O5 1:1 and 2:1, N1P1, and N2P1). 

Soil tillage was carried out to a depth of 0.4 m with a moldboard plow in October 2019, followed by a tandem disk harrow (0.1 m depth) to break clods. The organic fertilizer “Endurance N8” (Unimer s.p.a., Milano, Italy), containing organic nitrogen from animal protein source, and the organic soft rock phosphate “GAFSA 27” (Panfertil s.p.a, Ravenna, Italy) were used as N and P_2_O_5_ sources, respectively. On 27 November, 2019, both fertilizers were broadcasted on the soil surface, immediately followed by spike tooth harrow (0.05 m depth) to incorporate the fertilized and prepare the false seedbed. Weeds were uprooted to a depth of 0.05 m using a spring tine harrow on 7 and 17 December 2019. Winter wheat seeds were sown 90 kg ha^−1^ on 17 December 2019 with a row spacing of 0.13 m. Grain harvesting was performed at crop commercial maturity (grain moisture lower than 13%) on 15 July 2020.

### 4.2. Sampling and Samples Treatment

Sampling of bulk soil (thereafter indicated as soil) and rhizosphere soil was performed in May 2020. For each plot, five individual plants at flowering stage were selected, separated at least 5 m from each other. After digging up the entire plant (10 cm radius around the plant, and down to 20 cm soil depth) and shaking off the plant roots, the soil remaining attached to the roots was detached with a sterile scalpel and used as rhizosphere soil. Bulk soil was collected using a stainless steel sterile corer to sample the top 10 cm of soil in patches of bare soil between plants (ca. 30 cm apart from the plant) and sieved at 2 mm. Bulk and rhizosphere soil samples for each plot were combined to create composite bulk and rhizosphere soil samples. A total of 32 combined samples from 16 plots were analyzed (Table 2). Samples were stored at −80 °C until DNA extraction.

### 4.3. Soil Chemical and Physical Analyses

Soil pH was determined in a 1:5 soil-to-distilled water solution. Soil total nitrogen (STN) was determined by applying the dry combustion method on 5.0 mg soil samples, using a CHNS analyzer (CHN-S Flash E1112, Thermo-Finnigan LLC, San Jose, CA, USA) [48,49]. Soil samples (5.0 mg) were pretreated with chloride acid solution (10%) within silver cups to volatize the carbonate carbon and then analyzed by means CHNS to determine soil organic carbon (SOC) [50,51]. Ammonium nitrogen and nitrate nitrogen were determined by spectrophotometric determination (Lambda 20 spectrophotometer; PerkinElmer, Waltham, MA, USA) after extraction with the calcium chloride (CaCl_2_) procedure by Houba et al., (1995), which has the advantage of extraction uniformity for the considered N forms [52]. Aliquots of soil extract solution were treated according to the nitrate copper-cadmium reduction method [53], and the resulting solutions were spectrophotometrically analyzed at 540 nm after Griess reaction [54] for the determination of nitrate nitrogen. Further, aliquots of soil extract solution were treated following the Nessler method [55] and then absorbance analyzed at 420 nm for the determination of ammonium nitrogen. According to [56], soil samples were extracted with 0.5 M NaHCO_3_ at pH 8.5 to solubilize inorganic P forms which were spectrophotometrically analyzed at 882 nm after reaction with ammonium molybdate to determine the soil assimilable phosphorous (AP). Exchangeable cations (calcium, iron, magnesium, potassium, selenium, sodium, and zinc) were extracted from the soil using a 1 N NH_4_OAc extracting solution pH 7.0 and then determined using an ICP-OES analyzer (ICAP™ 7400 ICP-OES Analyzer, Thermo Scientific, Waltham, MA, USA). The soil cation exchange capacity (CEC) was determined using the BaCl2-triethanolamine method as proposed by Mehlich (1939) [57].

### 4.4. eDNA Extraction and 16S rRNA Gene Amplicon Sequencing

Environmental DNA (eDNA) was extracted from 500 mg of soil using a DNeasy PowerSoil Pro (Qiagen). From the extracted DNA, the bacterial V4 region of 16S rRNA genes was amplified with primers 515F (5′-GTGCCAGCMGCCGCGGTAA-3′) and 806R (5′-GACTACHVGGGTATCTAATCC-3′) [58] in a 25 µL total volume with KAPA HiFi HotStart ReadyMix, 1 µM of each primer, and 10 ng of template DNA with 25 cycles with the following temperature profile: 30 s at 95 °C, 30 s at 55 °C, and 30 s at 72 °C. PCR products were sequenced in a single run using Illumina MiSeq technology with pair-end sequencing strategy and a MiSeq Reagent Kit v3 (Illumina, USA). Library preparation (Nextera XT, Illumina, San Diego, CA, USA) and demultiplexing were performed following Illumina’s standard pipeline as previously reported [10].

### 4.5. Quantitative PCR of Genes Involved in the Nitrogen Cycle

Quantitative PCR was performed on eDNA using primers for the amplification of *nifH*, *amoA*, *nosZ*, *nirK* genes (primer sequences and citations are reported in Appendix A). Total bacterial load was estimated on the 16S rRNA gene. Reactions were performed with 5 ng of template DNA in 10 µL total volume on a QuantStudio7 Flexi apparatus (Applied Biosystems) with Sybr Green technology (Maxima SYBR Green qPCR Master Mix, Thermofisher). Reactions were performed in triplicates. Relative abundance of *nifH*, *amoA*, *nosZ*, *nirK* genes was expressed in relation to 16SrRNA, as previously reported [59,60].

### 4.6. Bioinformatic and Statistical Analyses

Illumina reads were trimmed using the “Trim Galore!” tool on Galaxy server (https://usegalaxy.org/, last accessed on 21 December 2021). Paired-end sequences were clustered into Amplicon Sequence Variants (ASVs) following the DADA2 pipeline (version 1.16) [61]. After filtering the sequences and removing the chimeras, the taxonomy assignment was carried out comparing our data against the SILVA NR99rel138 standard database of bacteria [62] using “DECIPHER” R package (version 2.18.1) [63] as implementation of DADA2 (SSU version 138 available at: http://www2.decipher.codes/Downloads.html, last accessed on 21 December 2021). Annotated ASVs count tables were processed in Phyloseq package in R environment version 4.0.5 [64].

The analysis of microbial communities was performed through the “Phyloseq” R package (version 1.34.0) [64]. For alpha diversity analysis, the Shannon and Simpson indices were calculated and plotted using the function “diversity()” within “microbiome” R package (version 1.12.0) [65]. Good’s coverage and Evenness indices were calculated through the R functions “goods()” and “evenness()”, respectively, within the “microbiome” R package (version 1.12.0). A Wilcoxon test for multiple comparison of averages was performed on alpha diversity indices using the “ggviolin()” and “stat_compare_means()” within the “ggpubr” R package (version 0.4.0). For ordination plots of phyloseq objects, a multivariate analysis based on Bray–Curtis distance and NMDS ordination generated using the “ordinate” function was performed, and plots were generated using the “plot_ordination()” function within phyloseq package. Rarefaction curves were generated using the “ggplot2” (version 3.3.3) [66] and “ranacapa” (version 0.1.0) [67] R packages using the “ggrare()” function on the phyloseq object. The “ggplot2” R package (version 3.3.3) was used to generate relative abundance plots.

Different community structures were analyzed using permutational multivariate analysis of variance (PERMANOVA) performed using the R packages “ggplot2” (version 3.3.3), “vegan” (version 2.5-7), and “pairwise.Adonis” (version 0.0.1) using the functions “adonis2()” and “pairwise.adonis()”, respectively. Clustering analysis for relationship discovery was performed using the “hclust()” R function selecting the agglomeration method “complete” on the distance matrix produced by the “dist()” function within the “stats” R package (version 4.0.5).

Statistical analyses for quantitative PCR data were performed in R environment (version 4.0.5) calculating the ratio between the threshold cycles of the target genes and 16S gene. Graphics of the ratio averages were performed using the “ggboxplot()” function within the “ggplot2′ R package (version 3.3.3). A Scott–Knott test was performed on averages using the functions “with()” and “SK()” within the “ScottKnott” R package (version 1.2-7) [68].

### 4.7. Prediction of Functional Abundances

The software PICRUSt2 (Phylogenetic Investigation of Communities by Reconstruction of Unobserved States, https://github.com/picrust/picrust2, last accessed on 21 December 2021) was utilized to investigate and predict the functional potential for each determined sequence in the microbial community. PICRUSt2 pipeline returned as output the metabolic pathway abundances, and functional gene profiling by comparison with MetaCyc and Kyoto Encyclopedia of Genes and Genomes databases [69]. Ranking of the most relevant pathways for varieties differentiation was conducted by running a SIMPER test using the function “simper()” with the “vegan” R package (version 2.5-7). For each contrast between varieties, the top three differentially expressed pathways were selected, and analyses were carried out using the function “metaMDS()” in the “vegan” R package (version 2.5-7) to perform nonmetric multidimensional scaling (nMDS) and the function “heatmap()” in the “stats” R package (version 4.0.5) to graphically represent results.

### 4.8. Links to Deposited Data

The sequences dataset was deposited in the SRA database under the BioProject PRJNA720503 (last accessed on 21 December 2021).

## Figures and Tables

**Figure 1 ijms-23-03616-f001:**
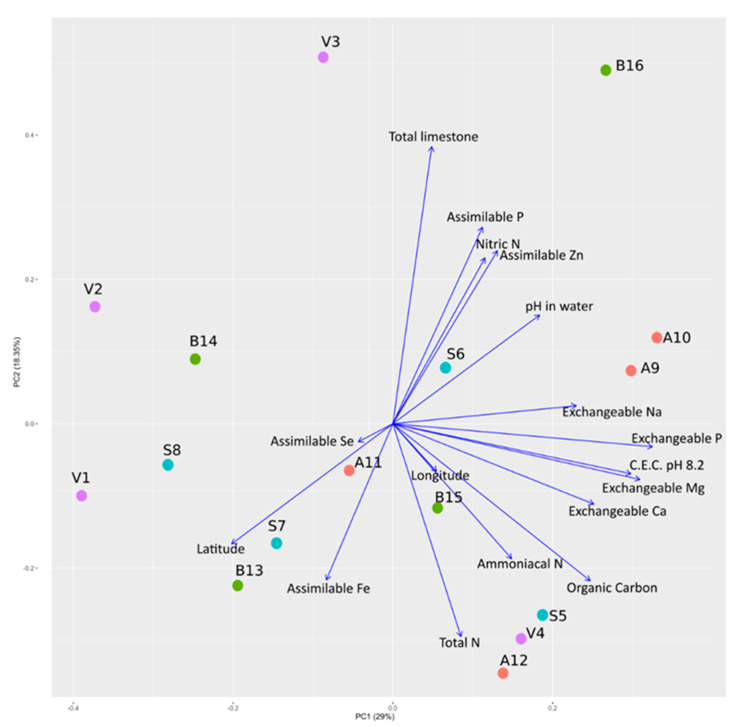
Influence of physicochemical composition and geographical coordinates of soil samples with respect to the sowed varieties. A Principal Component Analysis of physicochemical composition and geographical coordinates of soil samples is reported with a biplot showing the percentage of variance of the first two components. Codes of samples are as in Table 1.

**Figure 2 ijms-23-03616-f002:**
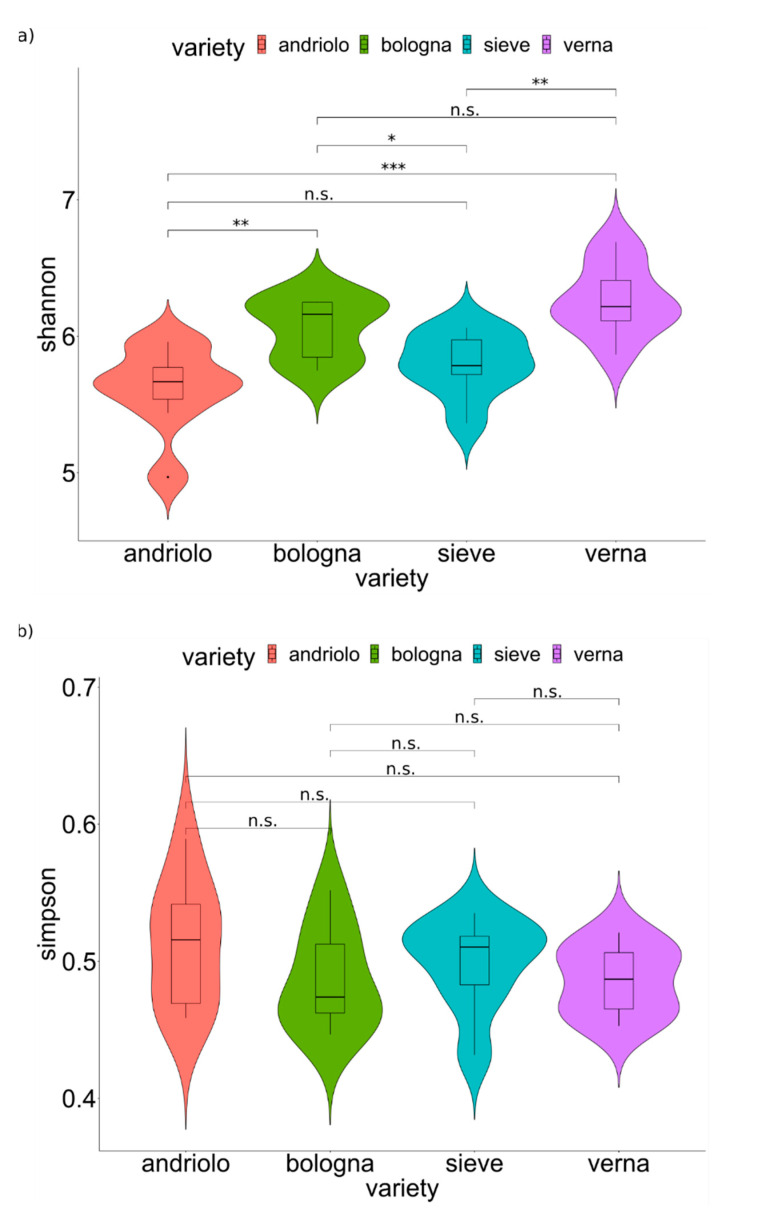
Diversity of soil and rhizosphere bacterial community. Values for alpha diversity indices are reported. Violin plots with samples (bulk and rhizosphere soil) are grouped according to varieties. Statistical analyses were performed using the Wilcoxon test (*p*-values are reported over the lines: *, *p* < 0.05; **, *p* < 0.01; ***, *p* < 0.001; n.s., not significant *p* > 0.05). The y axis indicates the values of the Shannon index (**a**) and the values of the Simpson index (**b**).

**Figure 3 ijms-23-03616-f003:**
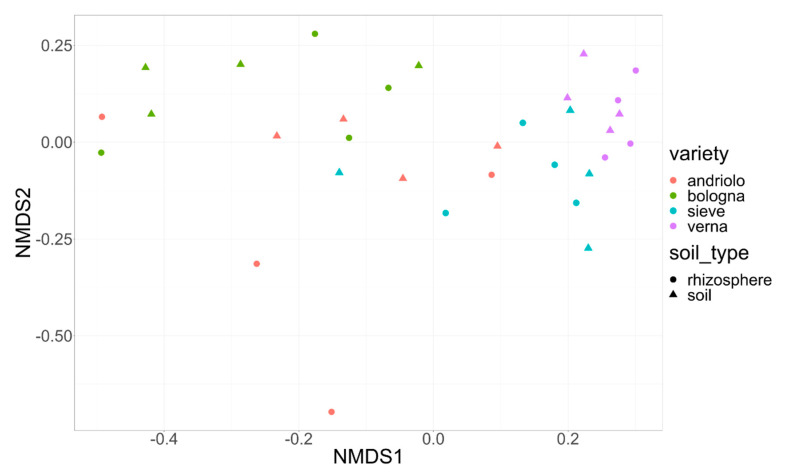
Soil sample distribution according to bacterial community. Nonmetric Multidimensional Scaling of soil and rhizosphere bacterial community. Ordination based on Bray–Curtis distance. Colors indicate the wheat variety, and shapes refer to the type of sample (circle = rhizosphere soil, triangle = bulk soil). Stress = 0.15.

**Figure 4 ijms-23-03616-f004:**
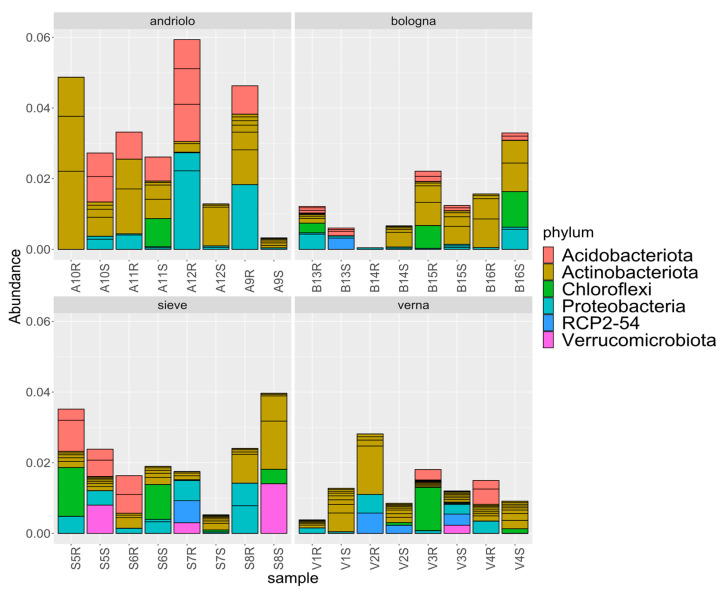
Relative abundances of taxonomies. Abundances of the top twenty detected ASVs are reported at phylum level in relation to variety.

**Figure 5 ijms-23-03616-f005:**
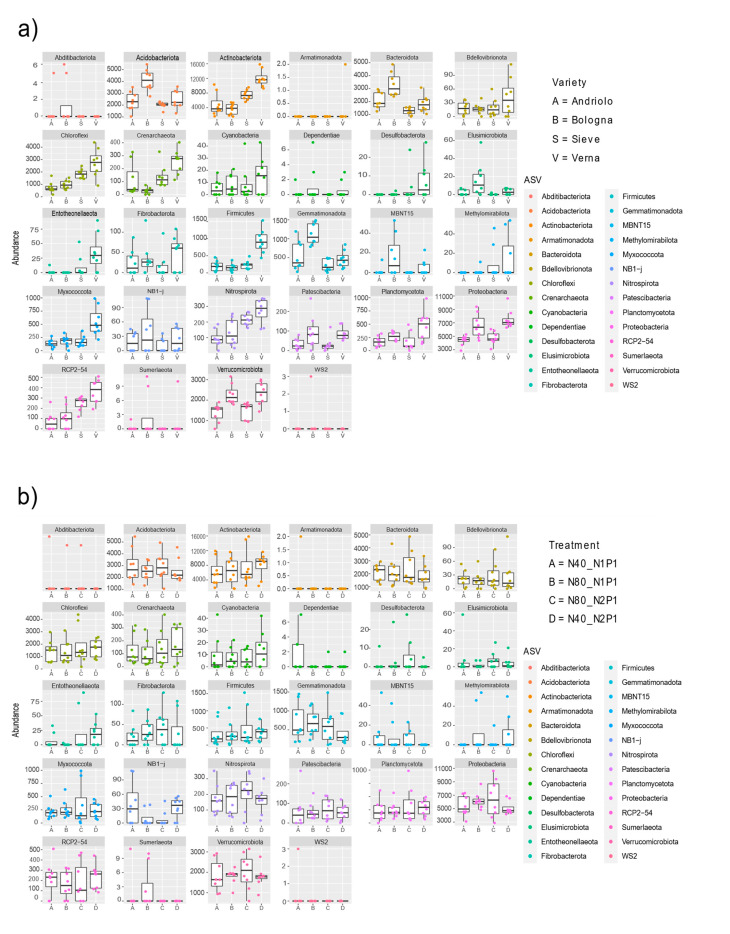
Variety and fertilization affect specific bacterial phyla abundances. Differential abundances of bacterial phyla in relation to wheat varieties and fertilization is reported from DeSeq2 analysis result. For each phylum, the abundance is reported on the y axis with respect to the varieties (**a**) and the fertilization (**b**).

**Figure 6 ijms-23-03616-f006:**
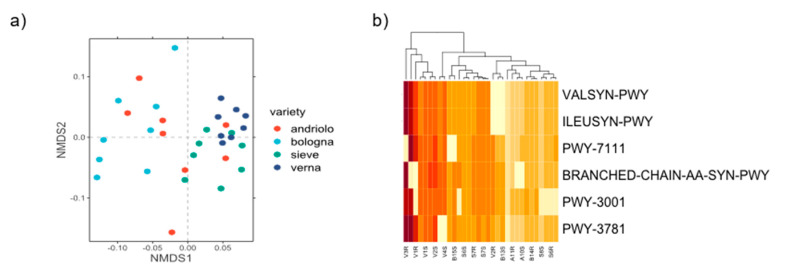
Clustering of pathways abundance. In panel (**a**), Nonmetric Multidimensional Scaling analysis (nMDS) was performed considering the predicted functional abundances of all identified pathways. In panel (**b**), a heatmap considering the top three pathways that differentiated each contrast between varieties returned by SIMPER test is represented. Codes of represented pathways correspond to the following pathway descriptions: L-valine biosynthesis (VALSYN-PWY), L-isoleucine biosynthesis I (from threonine) (ILEUSYN-PWY), pyruvate fermentation to isobutanol (PWY-7111), superpathway of L-leucine, L-valine, and L-isoleucine biosynthesis (BRANCHED-CHAIN-AA-SYN-PWY), L-isoleucine biosynthesis III (PWY-3001) and aerobic respiration I (cytochrome c) (PWY-3781).

**Figure 7 ijms-23-03616-f007:**
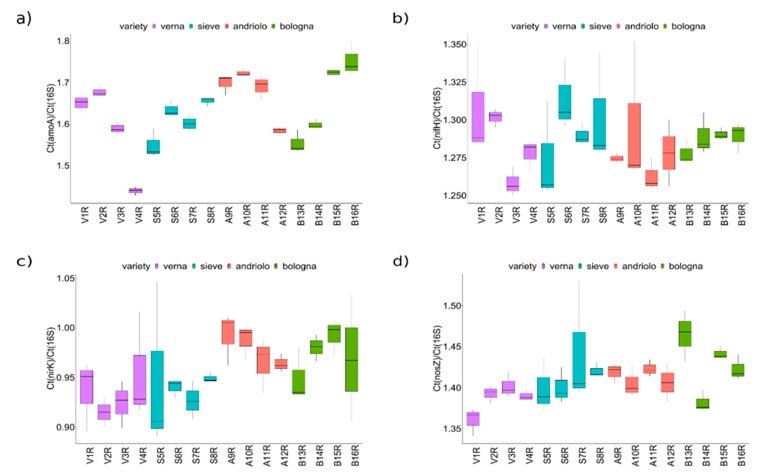
Abundance of genes involved in the nitrogen cycle. Plots report the averages of the relative gene quantification with respect to the 16S rRNA gene. Ratios of gene copies are a proxy of the abundances of bacteria harboring the indicated genes with respect to the total number of bacteria: (**a**) *amoA* gene; (**b**) *nifH* gene, (**c**) *nirK* gene, and (**d**) *nosZ* gene.

**Table 1 ijms-23-03616-t001:** Results of the soil physicochemical analysis. Sample codes refer to the name of the samples reported in Table 1. For each soil parcel analysis was performed on bulk soil (“S”) samples. Exc, exchangeable.

	Sample Codes															
	S5S	V2S	B14S	A11S	A12S	B13S	V1S	S6S	S7S	S8S	V4S	V3S	B16S	B15S	A9S	A10S
Organic C (g/kg)	0.9	0.7	0.8	1	1.1	1	0.8	0.9	0.8	0.8	1	0.7	0.9	1	1	1.1
Organic matter (g/kg)	1.6	1.3	1.4	1.7	1.8	1.7	1.4	1.6	1.4	1.4	1.7	1.2	1.5	1.7	1.8	1.9
Total N (g/kg)	1.1	1	1.1	1.1	1.2	1.2	1	1.1	1.3	1.1	1.2	1	1	1.2	1.1	1.2
Nitric N (mg/kg)	4	4	4	4	4	4	4	4	4	4	4	4	6	4	4	4
Ammoniacal N (mg/kg)	6	2	4	3	6.4	5	5	6.5	4	4	5	3	5.6	6	4	4
Assimilable P (mg/kg)	5	4	4	5	6	4	4	9	4	4	4	11	7	4	5	6
Exc. P (as K_2_O) (mg/kg)	569	316	321	381	407	378	313	487	378	376	607	362	559	436	591	607
Exc. P (mg/kg)	236	131	133	158	169	157	130	202	157	156	252	150	232	181	245	252
Total limestone (g/kg)	12	15	15	12	9	13	12	13	12	11	11	15	18	13	14	15
ExcCa (mg/kg)	3497	2768	2746	3279	3538	2798	2640	2962	2778	2594	3140	3111	2995	2918	3095	3116
Exc. Ca (as CaO) (mg/kg)	4895.8	3875.2	3844.4	4590.6	4953.2	3917.2	3696	4146.8	3889.2	3631.6	4396	4355.4	4193	4085.2	4333	4362.4
Exc. Mg (mg/kg)	153	96	84	119	114	106	88	112	104	94	147	75	155	134	163	149
Exc. Mg (as MgO) (mg/kg)	255	160	140	198	190	177	147	187	173	157	245	125	258	223	272	248
Exc. Na (mg/kg)	18	13	16	13	17	15	15	19	19	19	18	16	20	17	21	19
Assimilable Fe (mg/kg)	12.5	11	11	11	9.1	12	12	9.3	11	10	12	10	8.5	10	11	11
Assimilable Zn (mg/kg)	0.4	0.3	0.4	0.4	0.3	0.4	0.3	0.6	0.5	0.4	0.5	1	0.5	0.6	0.5	0.8
Assimilable Se (mg/kg)	0.3	0.2	0.1	0.2	0.1	0.1	0.1	0.1	0.2	0.3	0.1	0.1	0.2	0.1	0.1	0.1
pH in water	8.2	8.2	8.2	8.2	8.2	8.1	8	8.2	8.2	8.2	8.1	8.2	8.2	8.1	8.3	8.3
C.E.C. pH 8.2 (meq/100 g)	19	15	15	18	19	15	14	16	15	14	18	17	17	16	18	18

**Table 2 ijms-23-03616-t002:** Samples analyzed. The code, wheat variety, the fertilization of the plot, and their geographical position is reported. Numbers in the sample code refer to the plots. “S” and “R” in the sample code indicate soil and rhizosphere fraction, respectively.

Variety	Code	Nitrogen Fertilization (N kg ha^−1^ yr^−1^)	Phosphorous Fertilization (P_2_O_5_ kg ha^−1^ yr^−1^)	Latitude (N)	Longitude (E)
Verna	V1S	40	80	43.051129°	11.690833°
	V1R	40	80	43.051129°	11.690833°
	V2S	80	160	43.050959°	11.691126°
	V2R	80	160	43.050959°	11.691126°
	V3S	80	80	43.050755°	11.691438°
	V3R	80	80	43.050755°	11.691438°
	V4S	40	40	43.050547°	11.691890°
	V4R	40	40	43.050547°	11.691890°
Sieve	S5S	40	40	43.050953°	11.692320°
	S5R	40	40	43.050953°	11.692320°
	S6S	80	80	43.051126°	11.691964°
	S6R	80	80	43.051126°	11.691964°
	S7S	80	160	43.051343°	11.691626°
	S7R	80	160	43.051343°	11.691626°
	S8S	40	80	43.051525°	11.691219°
	S8R	40	80	43.051525°	11.691219°
Andriolo	A9S	40	40	43.049740°	11.690994°
	A9R	40	40	43.049740°	11.690994°
	A10S	80	80	43.049901°	11.690666°
	A10R	80	80	43.049901°	11.690666°
	A11S	80	160	43.050036°	11.690376°
	A11R	80	160	43.050036°	11.690376°
	A12S	40	80	43.050180°	11.690020°
	A12R	40	80	43.050180°	11.690020°
Bologna	B13S	40	80	43.050763°	11.690400°
	B13R	40	80	43.050763°	11.690400°
	B14S	80	160	43.050531°	11.690761°
	B14R	80	160	43.050531°	11.690761°
	B15S	80	80	43.050347°	11.691104°
	B15R	80	80	43.050347°	11.691104°
	B16S	40	40	43.050193°	11.691440°
	B16R	40	40	43.050193°	11.691440°

## Data Availability

Data are available at https://www.ncbi.nlm.nih.gov/bioproject/?term=PRJNA720503, accessed on 1 February 2022.

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
