# Peer review of "Differential Response of Wheat Rhizosphere Bacterial Community to Plant Variety and Fertilization"

_ijms, 2022, doi:10.3390/ijms23073616_

Round 1

Reviewer 1 Report

The manuscript of Cangioli and coauthors is an experimental study aimed at clarifying the influence of fertlization and genotype (variety) on the structure and taxonomy of bacterial community recruited by plants in the rhizosphere. The study was conducted under field conditions, a valuable aspect of the research. The topic is of interest and the experimental plan is good, however there are few issues that have to be addressed in the manuscript before consider it suitable for publication.

Main comments:

The authors analyzed bulk and rhizospheric soils, however the analysis of bacterial community structure does not divide the samples according to this categorzation, that is quite strange. The authors should clearly discuss the limitation of the adopted sampling, in the light of obtained results.

The Discussion section has to be rewritten. It seems largely a repetition of the results while the relation between the presented data and the literature should be more clearly outlined.

Specific comments:

Abstract

l. 22: typo ‘metabolism’

l. 21-24: long sentence that could be reformulated to improve readability and avoid repetition

Introduction:

l.58 ‘but at the expense of the reduction of the concentration of vitamins, and micronutrients in grains’ could be for example ‘but also implied a lower vitamin and micronutrient concentrations in grains’

l. 63-64: the authors stated ‘These features suggest that ‘old’ and ‘modern’ wheat varieties may harbor different root-associated microbiota related to the different nutrient requirements and root systems’. I agree but the long paragraph above this sentence does not refer explicity to the root system changes caused by breeding while it focused on other difference between old and new varieties (l. 45-62). The authors could consider to shorten a bit this part and/or use these lines to better highlight the differences about nutrient requirements (clearly expained) and root systems.

l.65: please explain the meaning of ‘more diverse’, characterized by higher alpha-diversity indices? Or from a taxonomical perpsective?

l. 71: ‘Given the resurged importance of old wheat varieties..’, such importance remains unknown for the reader, is it related to climate changes or other factors?

Materials and Methods

l. 273: ‘The 0-40 cm layer’ I suppose the authors focused on the topsoil due to the fact it is the more interested by root system growth and also by agronomic practices like tillage, maybe they could better specify about it.

l. 277: were previous crop residue mixed into soil as manure?

l. 281-284: I am not sure all details about the origin of the germoplasm are essential in the manuscript, considering that the important aspect is clarified (germoplasms are different) and suitable references are provided. Alternatively the could be provided as Table 1 so the readers when look at the results (l.87) do not wonder, for instance, what is Andriolo.

l. 308: typo, missing dot

l. 308: ‘For each plot’: 16 plots as the 16 treatements? (above there is also the reference to subplots and sub-subplots so I got lost. I see then below you make the summary of sample number, so I suggest to simply state al l. 308 that each plot corresponds to a different treatment

l. 318 parcel could be replaced by plot, for consistency of terms

l. 348: please indicate DNA quantity used in the PCR reaction, and do the same for the qPCR protocol

l. 356: please state in this subsection that a supplementary table is included reporting each primer name and sequence

l. 403 typo ‘the The’

Results

Table 1. Please pay attention to formatting that can be improved. Pay attention also to typo (for example, a missing traslation from Italian, come instead of as). Personally I would avoid all these ‘as…’ that can be specified in the methods. Also, authors could use some abbreviation and specify them in the legend (i.e., Exchangeable can be Exc.) and the name of elements could be replaced by the symbols.

l.85, in the Materials and Methods you defined the bulk soil simply as soil, then throughout the results I see the term bulk soil. Personally I do prefer the ‘bulk soil’ option, so I suggest to correct the Materials and Method accordingly.

l. 86: soil physicochemical parameters and the location of sample collection (i.e., latitude and longitude of the plots)

Figure 1: legend is misleading. Remove the reference to microbiota if te PCA has been performed on soil physicochemical parameters and the location of sample collection. Moreover in the text the authors stated that Andriolo samples were influenced by total N and Exch. K buti t is difficult for the readers to get the information out of the Figure. Please consider to prepare the Figure using different color for the dots according to plots.

l. 108: ‘Varieties differed in relation to Shannon Index’. Please provide a p value for this statement if you applied a main-test permanova and also provide a more accurate description of the presented results, considering the statistical pairwise comparison included in Figure 2

l. 119-130: I would use the term taxonomic diversity, it is better to indicate the results of Fig 3 as beta-diversity

Figure 3. Similarities among microbiota, it should be something like: Soil sample distribution according to bacterial community composition

l.132 and Fig.4 legend: relative, not absolute abundance

l. 143: ‘as well as several or-143 ders of Actinobacteriota’ is a bit too hurried

l. 132: why the authors decided to show only the top 20 ASVs detected? Moreover, it is important to define what they mean as ‘top 20’ and how they were identified, which is the range of relative abundance of these ASVs over the total bactermial communities of the sample. Avoid vague indications.

l. 135, RCP2-54 is not indicated in the text while it is present in fig.4. Even though it is a candidate divisioni t cannot be neglected from results.

Figure 5: pay attention, it is indicated ‘OTU’ instead of ‘ASV’. I suggest to improve also the written part of the fugure (axis title, legends, etc). Moreover, here instead to use the letters A-B-C-D the authors could use the code explain in the Materials and Methods (N40-N80-N1P1-N2P2) otherwise I think those codes are useless (unless they are used in other part of the manuscript and I did not find them).

l. 155: In the section ‘Prediction of potential functions and metabolic pathways’ please provide information on the predicted metabolic pathways and enzymes involved in the N cycle. Given the focus on this biogeochemical cycle in the study, I think that it could be interesting to see if there are other (inferred) information on it.

l. 156: replace microbiota by bacterial community (please check also throughout the manuscript)

l. 159: ‘showed a clustering’, replace showed by suggested unless statistic analysis has been performed

l. 170, Figure 6 legend: typo

l. 184: abundance of the total bacterial community. Concerning the qPCR analyses on the genes involved in the N cycle I think it would have been interesting to see also a comparison with the bulk soil samples, that could be seen as a reference. Why the authors did not include this sample in the qPCR analyses? I think this is a major critical point in the manuscript: in fact, without the data on bulk soil gene abundance how the authors can exclude the role of ‘fertilization’ instead of the ‘variety’ factor?

Discussion

l. 210-211: unclear, I think tha authors should better discussed the fact they could not distinguish bulk vs rhizospheric soils. If during the sampling they were not able to catch the real bulk soil fraction the reasons should be better outlined and stated. In this perspective, I can understand the absence of qPCR data on N cycle genes

l. 212 here is the only point where a variety name is reported as: “Bologna” instead of Bologna.

Reviewer 2 Report

The manuscript reports a differential response of wheat rhizosphere bacterial community composition and function to plant variety and different NP fertilization regimes. The methodology involved 16S rRNA gene amplicon sequence analysis and quantitative PCR analysis of main functional bacterial genes involved in the nitrogen cycle. Regarding plant variety, four Triticum aestivum, L. varieties were used, namely three “old” genotypes (Andriolo, Sieve, and Verna) and a modern variety (Bologna). The results showed a variety-specific response of rhizosphere microbiota in terms of community structure, composition, and metabolism, with fertilization playing a negligible role.

Although the manuscript reports an exhaustive work with many results to explore, I found that these were not sufficiently discussed. The authors refer in the abstract and introduction the correlation between plant varieties and the ability of plant recruitment, with consequences to rhizosphere microbiota composition. For instance, they refer “Crop breeding, especially after the massive use of nitrogen fertilizers, which led to varieties responding better to nitrogen fertilization has implicitly modified the ability of the plant root to recruit an effective microbiota…”,  and in the "Introduction “ section they mention previous studies, where “..bacterial communities in the rhizosphere of ‘old’ wheat varieties were found to be more diverse than that detected for modern wheat…” . However, differences found for the modern Bologna variety or no differences (frequently grouped with the Andriolo variety) are not discussed in this perspective. An enrichment of the discussion regarding this aspect, and in comparison to previous studies, is necessary. In addition, as this manuscript is proposed to be part of the special issue “Plant–Microbe Interactions”, willing to address “questions including the molecular basis of plant–microbe interactions and the evolution of plant–microbe interactions”, an evolutionary perspective on the differences found in microbiota composition and gene function for each wheat variety should be attempted in the discussion. Other interesting points, as the presence of RCP2-54 uncultured phylum among the top twenty detected amplicon sequence variants, are also not discussed.

Below, the proposed changes are detailed (per section).

Abstract:

line 15- insert “of” in the sentence: “…in the microbiota of a varietal effect of N and P fertilization treatment.”

 line 22- replace “aerobic metaoblisms” by aerobic metabolism.

Introduction:

 lines 54 and 55- insert “the first” in the sentence: “Therefore, ‘old’ wheat varieties were rapidly replaced by the ‘modern’ ones, the first being very tall and poorly productive,…”

 lines 73 and 74- insert “of” in the sentence: “…to investigate the presence of a differential effect of the same N and P fertilization treatment…”

Results:

line 86- in the sentence “Analysis (PCA) was performed on the soil parameters (including latitude and longitude of the plots).” Remove brackets and change the sentence to “…soil parameters and included latitude and longitude of the plots.”

lines 91 and 92- in the table title, the sentences “Sample codes refer to the name of the samples reported in Table 2. For each soil parcel, analysis was performed on bulk soil (“S”) samples.” should be moved from the title and placed as a footnote. Also, correct the sentences as suggested. 

In the table 1, replace “Exchangeable Magnesium (come MgO) (mg/kg)” by exchangeable Mg (as MgO).

Figure 1- in the figure, the name of physicochemical parameters is cut. If possible, increase the size of axis titles.

lines 94-96- change the legend of figure 1: “Influence of physicochemical composition and geographical coordinates of soil samples with respect to the sowed varieties. A Principal Component Analysis of physicochemical composition and geographical coordinates of soil samples is reported with a biplot showing the percentage of variance of the first two components.”, i.e., remove “is shown” from the end of the sentence.

 Figure 5b)- the values in B and D of 160 KgP2O5 and 160 Kg N are not in agreement with the reported in the “Materials and Methods” section or to the values reported in supplementary materials, for instance in Table S4.

line 155- remove “then” in the sentence “Functional profiles of the microbiota were (then) inferred.”

lines 156 and 157- insert “the” in the sentence “All the identified metabolic pathways present in the taxa retrieved in the microbiota are listed in the Supplementary Table S6.”

lines 166 and 167- correct the sentence as follows: “Interestingly, this subset of pathways also showed a clustering related to varieties and in particular Verna grouped apart from the other three varieties (Figure 6b).”

lines 172 and 173- change the sentence as follows: “In panel b) a heatmap considering the top three pathways that differentiate each contrast between varieties returned by SIMPER test is represented.”

line 182- replace NH4+ by NH3.

Discussion:

lines 198—212- “Crop breeding, especially after the massive use of nitrogen fertilizers, which led to varieties responding better to nitrogen fertilization [36], has implicitly decreased the pressure toward plant genotypes highly efficient in the use of native soil nutritional resources, then indirectly reducing the ability of the plant root to recruit an effective microbiota [7,8,37]. …The three old varieties and the modern “Bologna” variety showed a differential abundance of several microbial taxa”.  The authors could hypothesize on the taxa and functions found for the rhizosphere microbiota of the modern variety bologna in comparison to the other varieties and if they consider that a reduced or simply a different ability exist in the recruitment of effective microbiota. For instance, regarding the quantitative analysis of the nitrogen cycle genes, with Andriolo and Bologna showing higher abundance of nirK and amoA genes, the ability of recruitment of important bacteria in nitrogen cycle would be enhanced, but this is not discussed.

line 228- at line 228, it would be also interesting to discuss the presence of RCP2-54 uncultured phylum within Binatota in bologna, sieve and verna. Murphy et al. (10.21203/rs.3.rs-79662/v1) found that “all orders in the Binatota encoded the capacity for aerobic methylotrophy, being methylamine oxidation related to nitrogen metabolism.

line 262- correct the sentence as follows: “It would be highly relevant, in the perspective of a low-input precision agriculture to clarify the extent and impact on plant productivity of a variety- fertilization interaction and to exploit the potentialities of variety-specific microbiota.”

Materials and Methods:

line 278- Triticum aestivum in italics

line 303- use brackets in “Winter wheat seeds were sown (90 kgha-1) …”

line 308- add a full stop after “… in May 2020.

line 324- “…using a CHNS analyzer …”

line 326- “…the carbonate carbon and then analysed by means of CHNS to determine the soil organic carbon (SOC) …”

line 329- “after extraction with the calcium chloride (CaCl2) …”

line 330- “..procedure described by Houba et al. (1995) …”

line 336- “... with 0.5 M NaHCO3 at …”

line 337- introduce a comma in the sentence: “…solubilize inorganic P forms, which were…”

line 340- as above, correct the formula in the sentence: “…were extracted from the soil using 1 N NH4OAc…”

line 343- Correct the formula in the sentence: “…using the BaCl2-triethanolamine method …”

line 356- “Quantitative PCR was performed on eDNA using primers

line 403 and 404- Please clarify the text in these lines: “PICRUSt2 pipeline returns as output the The metabolic pathway abundances and …”

Supplementary Materials

line 431, Figure S2- There is no bar plot in relation to a) soil type, so correct the sentence as follows: “Bar plots showing the relative abundances of bacterial phyla grouped by N and P treatment in relation to wheat varieties;”
